# Consensus for Flow Cytometry Clinical Report on Multiple Myeloma: A Multicenter Harmonization Process Merging Laboratory Experience and Clinical Needs

**DOI:** 10.3390/cancers15072060

**Published:** 2023-03-30

**Authors:** Iole Cordone, Rachele Amodeo, Silvia Bellesi, Fiorella Bottan, Francesco Buccisano, Maria Stefania De Propris, Serena Masi, Valentina Panichi, Maria Cristina Scerpa, Ombretta Annibali, Velia Bongarzoni, Tommaso Caravita di Toritto, Ugo Coppetelli, Luca Cupelli, Paolo de Fabritiis, Luca Franceschini, Mariagrazia Garzia, Alessia Fiorini, Giacinto Laverde, Andrea Mengarelli, Tommaso Za, Maria Teresa Petrucci

**Affiliations:** 1Department of Research, Advanced Diagnostic and Technological Innovation, IRCCS Regina Elena National Cancer Institute, 00144 Rome, Italy; 2Clinical Pathology and Biochemistry Unit, Sant’Andrea University Hospital, 00189 Rome, Italy; 3Department of Diagnostic Imaging, Oncological Radiotherapy and Haematology, IRCCS Foundation A. Gemelli University Hospital, 00168 Rome, Italy; 4Clinical Pathology Unit, San Giovanni Addolorata Hospital, 00184 Rome, Italy; 5Haematology and Stem Cell Transplant Unit, Department of Biomedicine and Prevention, University of Rome ‘Tor Vergata’, 00133 Rome, Italy; 6Immunophenotype Laboratory, Department of Translational and Precision Medicine, ‘Sapienza’ University, 00185 Rome, Italy; 7Microbiology and Virology Unit, Department of Oncology and Haematology, Belcolle Central Hospital, 01100 Viterbo, Italy; 8Haematology and Stem Cell Unit, Santa Maria Goretti Hospital, ASL Latina, 04100 Latina, Italy; 9Haematology and Stem Cell Transplant Unit, Campus Bio-Medico University, 00128 Rome, Italy; 10Haematology Unit, San Giovanni Addolorata Hospital, 00184 Rome, Italy; 11Haematology Unit, ASL Roma 1, 00193 Rome, Italy; 12Haematology Unit, Sant’Eugenio Hospital, ASL Roma 2, 00144 Rome, Italy; 13Haematology and Stem Cell Transplant Unit, San Camillo Forlanini Hospital, 00152 Rome, Italy; 14Department of Oncology and Haematology, Belcolle Central Hospital, 01100 Viterbo, Italy; 15Haematology Unit, Sant’Andrea University Hospital, 00189 Rome, Italy; 16Department of Research and Clinical Oncology, IRCCS Regina Elena National Cancer Institute, 00144 Rome, Italy; 17Haematology Unit, Department of Translational and Precision Medicine, ‘Sapienza’ University, 00185 Rome, Italy

**Keywords:** multiple myeloma, flow cytometry, clinical report, minimal residual disease

## Abstract

**Simple Summary:**

We report an Italian multicentre study with the aim to reach a consensus based on the essential data to be included in the flow cytometry clinical report of patients with MM at diagnosis and after treatment. From the pre-analytical phase, through sample processing, data acquisition, analysis and evaluation of the potential limitations and pitfalls of the entire process, the study reaches a final conclusion shared by laboratories and clinicians according to the most up-to-date principles and recommendations.

**Abstract:**

Flow cytometry is a highly sensitive and specific approach for discriminating between normal and clonal plasma cells in multiple myeloma. Uniform response criteria after treatment have been established by the International Myeloma Working Group and the EuroFlow Group; however, the way in which flow cytometry data are reported has suffered from no collaborative or multicentre efforts. This study, involving 8 expert laboratories and 12 clinical hematology units of the Lazio region in Italy, aims to produce a uniform and shared report among the various Centres. From the pre-analytical phase to sample processing, data acquisition, analysis, and evaluation of the potential limitations and pitfalls of the entire process, the study reaches a final conclusion shared by laboratories and clinicians according to the most updated principles and recommendations. The aim was to identify the necessary data to be included in the clinical report by using multiple-choice questionnaires at every single stage of the process. An agreement of more than 75% of the laboratories was considered mandatory for the data to be included in the report. By ensuring the operational autonomy of each laboratory, this study provides a clear report that limits subjective interpretations and highlights possible bias in the process, better supporting clinical decision-making.

## 1. Introduction

Flow cytometry characterization is a key technique in the diagnosis, classification, and disease monitoring in hematological diseases [1]. More recently, the study of minimal/measurable residual disease (MRD) has assumed primary importance in post-treatment monitoring studies by exploiting the high sensitivity and specificity of phenotypic analysis in rare cell identification and is considered a major prognostic factor able to surrogate survival analysis in several hematologic diseases [2,3,4,5]. Thereafter, cytometry laboratories have faced a dramatic increase in the requests for flow-MRD identification.

The availability of an increasing number of therapeutic options for the treatment of multiple myeloma (MM) patients is significantly changing the natural history of the disease. This, in turn, has made it necessary to define sensitive and specific tools to assess treatment response through accurate identification and monitoring of the monoclonal plasma cell (PC) population [6]. Therefore, in the past few years, a number of studies have been published focusing on the clinical relevance of MRD assessment by flow cytometry, concluding that the achievement of MRD negativity is a relevant predictor of clinical outcome [7,8,9,10]. However, the phenotypic features of clonal PCs and the irregular, patchy distribution within the bone marrow represent both a valuable opportunity and a technical challenge to evaluate the efficacy and reproducibility of flow-MRD in MM [11]. Outside of clinical trials, one of the major problems is represented by the consistency of results among different laboratories. Common causes of discrepancies are linked, but not limited, to different procedures, panel design, instrumentation and analysis software.

From the analysis of the most updated principles and recommendations of the Next Generation Flow-MRD process, we report an Italian multicentre study that aims to reach a consensus based on essential information to be reported in the flow cytometry clinical report of patients with MM at diagnosis and after treatment.

## 2. Methods

### Participants and Screening Policy

The project was joined by 12 hematological centers related to 8 onco-hematological diagnostic laboratories of the Italian Multiple Myeloma Lazio Group (Appendix A). The minimum requirement for participation in the study was the use of an 8-color flow cytometer, regardless of the type of instrument, reagents, and panel design. Participating laboratories were proficient in staining, data analysis, gating strategies, and data interpretation for MM clinical diagnosis according to the practice guidelines published by the International Council for Standardization in Haematology (ICSH), the International Clinical Cytometry (ICC), the European Myeloma Net guidelines and the EuroFlow Group [12,13,14,15]. In addition, 6/8 laboratories also had experience in MM flow-MRD studies.

A comparison between the 8 laboratories regarding the strategies of phenotypic analysis, as well as an agreement on the data that must be included in the final report, was reached through the use of a questionnaire; yes (Y), no (N) or not relevant (NR) were the 3 possible answers. The discordant results were discussed collectively. The questionnaire was divided into 5 macro-areas: (1) patients’ personal information, clinical data and biological material; (2) methods of cytometry; (3) quality of the sample; (4) qualitative and quantitative results of flow cytometry analysis; (5) final comment/conclusion of the report. Table 1, Table 2, Table 3, Table 4 and Table 5 show survey conclusions. An agreement of ≥75% between the experts was required to include data in the clinical report. The final report was submitted to and approved by the clinicians of the regional Myeloma Group.

## 3. Results

### 3.1. Patient Personal Information, Clinical Data and Biological Material

Table 1 shows the answers to the questionnaire concerning patient information, clinical data, and the biological material, together with additional information about the timing of processing. Most items #1 to #17 were approved while the following 4 points were addressed: (a) type of anticoagulant (#10), (b) amount of biological material (#11), (c) first pool (#12), and (d) cellularity (#13).

The majority deems the type of anticoagulant irrelevant to report on since it is generally recorded in the laboratory worksheet. As a quality control of the sample, it was agreed (75%) to report the amount of blood (ml) collected for the flow cytometry study. Additionally, the first pool (ideally 3–5 mL), considered the sample of choice for MRD studies to avoid peripheral blood contamination of the bone marrow sample [16,17], has been accepted for reporting and must be provided by the clinician. As for cellularity, it is advisable to report the absolute number of white blood cells in the sample. Additionally, the importance of reporting sample processing times has been highlighted, which is of great potential relevance when the sample is sent to external reference laboratories. Among the clinical information, all participants agreed on the need to notify the laboratory of treatment with anti-CD38 antibodies (#17).

### 3.2. Cytometry Analysis

Table 2 focuses on the information regarding the methods used and the number of events analyzed for the flow cytometry assessment of MM. It was considered not relevant to report whether bulk lysis is performed (#18), as well as the type of permeabilizing reagents used for intra-cytoplasmic staining (#19). Instead, the report must include the gating strategy applied to identify the different cell populations (#20), the number of total events acquired, and the total number of PCs acquired and analyzed (#21–22).

### 3.3. Quality Control of the Sample: Cytometric Myelogram

Table 3 shows the strategies to identify the different bone marrow cells and their relative distribution. For this purpose, the use of all markers included in the phenotypic panel is recommended. Taking into consideration the possible traps in MM characterization and analysis, a list of ‘warning/potential pitfalls’ has also been discussed.

As for nucleated red blood cells (NRBC, #23–24), it is recommended to evaluate the percentage of erythroblasts as the CD45^neg^ CD138^neg^ and/or CD38^neg^ events, being the majority of clonal PC CD45-weak or -negative (Figure 1a). Overestimation of the amount of NRBC can be done by gaiting CD45^neg^/SSC^low,^ potentially including small-size PCs (#24) (Figure 1b). The NRBC value obtained from the white blood cell count performed by a hematology analyzer (#25) has been considered not relevant.

Lymphocytes and hematogones are identified through a standard gating strategy based on MM panel markers (Figure 1b) (#26–27), and the lymphocyte percentage should be reported. The enumeration of B-cell precursors (CD19^+^ CD38/CD81^bright^) (#28), normally absent in peripheral blood, may be helpful in assessing hemodilution. Myeloid precursors and mast cells are evaluated and reported by the use of CD117 (Figure 1c,d), a key marker in MM characterization related to the clonal PC population in 35% of cases [18]. Warning: a gate performed on the CD117/CD45 positive population includes PCs expressing these markers while a CD117^+^/CD138^neg^ gate excludes CD117 positive PCs (#30–31). An increase in the percentage of CD45/CD117^+^ myelocytes can be considered a potential index of myelodysplasia (#32). Mast cells (#33) are a bone-marrow-specific cell type. Their absence (<0.002%) can suggest hemodilution.

Regardless of the strategy of analysis used, the monocyte percentage must be reported (#34–36). Despite the possible correlation between the identification of CD56^+^ monocytes and a state of myelodysplasia in MM (#37) [19], only 50% of the participants consider this data useful to report.

Due to the frequent lack/weak CD45 expression in the clonal population, the PC percentage must be evaluated on the entire cell population of the sample and not on the CD45^+^ leukocytes (#38–42). PCs are recognized by CD38/CD138^bright^ co-expression (Figure 1f); however, a possible CD38 and/or CD138^dim-absent^ PC expression must always be taken into consideration; thereafter, a gate on CD38 or CD138^bright^ populations (Figure 2a,b) will exclude the clonal population from the analysis of these rare MMs (#38–41). Moreover, a gate on CD38^bright^ PCs can lead to false negative results after anti-CD38 therapy (#39); warning: a population of CD38-negative monocytes or the absence of B-cell precursors (CD19/CD38 lymphocytes) may raise attention to a possible anti-CD38 treatment that has not been notified to the laboratory (#39). Several markers, different from the standard diagnostic monoclonal anti-CD38 antibodies, must be considered to identify the PC population after anti-CD38 treatment. All participants recommend reporting whether markers other than CD38 and CD138 are utilized for PC identification (#42).

Overall, in line with EuroFlow recommendations, it was agreed to include in the report a cytometric myelogram represented by the following parameters: (a) % NRBC (CD45^neg^/CD138^neg^); (b) % lymphocytes (CD45^++^/SSC^low^); (c) % myelocytes (CD45^+^/SSC^hi^); (d) % myeloid precursors (CD45/CD117^+^); (e) % mast cells (CD117^hi^/SSC^hi^); (f) % monocytes (CD45/SSC^int^ or CD38^+^/SSC^int^); (g) % PCs (CD38 and/or CD138^++^).

### 3.4. Flow Cytometry Analysis of PC and Lymphoid Populations

Table 4 describes the agreement on qualitative and quantitative results of the PC and lymphocyte evaluation. According to the EuroFlow guidelines [20], the phenotypic panel includes the following markers: CD19, CD20, CD27, CD28, CD38, CD45, CD56, CD81, CD117, CD138, cyto Ig-Kappa, and cyto Ig-Lambda. The strategy of analysis was discussed with particular attention to evaluating and highlighting possible sub-clones within the PC population. The role of Ig light chains’ expression/restriction on all PC and lymphocyte subpopulations was deeply discussed, and results are presented below.

Participants agreed on reporting the list of antibodies tested (#43), as well as the percentage of PC out of the total cellularity (#44). Regarding their analysis, the percentage of positive cells for each marker can be expressed as positive cells on total cellularity (#45–46) or within the PC compartment (#47–48). The majority strongly suggest reporting the percentage of both clonal and normal PC within the whole PC population. By contrast, only 62% of the laboratories agreed on including the *ratio* obtained from the percentage of clonal cells on total PC in the clinical report (#49). CD38 and CD138 are the backbones in MM characterization: all centers recommended specifying if other markers are utilized for the PC identification (#50–53). Moreover, the analysis strategy performed on both PCs and the lymphoid population was shared. The choice of the combinations, fluorochromes, and analysis system was left to the individual laboratories. Markers such as CD19, CD20, CD45, CD56, CD117, cyto Ig-Kappa, and cyto Ig-Lambda were selected as mandatory. By contrast, CD27, CD28, and CD81 were left to the choice of the individual laboratory (#54-63). All participants agreed to consider the Ig light chain Kappa/Lambda *ratio* on PC subpopulations a key marker to discriminate between clonal *versus* normal PCs and that the Ig light chains *ratio* on PC (pathological Kappa/Lambda *ratio* <0.5 or >4.0) must be reported. It is recommended to evaluate Kappa/Lambda expression on each informative antigen: both on the markers that identify a normal population (e.g., CD19/CD45^+^ PCs) and on the surface aberrant marker (SAM), alone or in positive and negative combination with the ‘normal’ markers (#64–73). As an internal control, CD45/CD19^+^ PCs represent, in most cases, a polyclonal population, as only a minority of myelomas are CD19^+^ (Figure 2). Additionally, the majority agreed to include in the report only the percentage of CD19 and/or CD38^+^ and CD20^+^ B lymphocytes. The other lymphoid markers, even informative, should not be included in the report (#74–85).

### 3.5. Final Comment/Conclusion of Report

Table 5 shows the consensus on the comment and conclusion of the report. It must clearly state if clonal PCs are present or absent, describe the list of antigens tested and the aberrant phenotype of the neoplastic PC population, declare if the sample was adequate or not for the study, the maximum sensitivity reached, the MRD status (to date minimal residual positivity is defined as an accumulation of ≥50 monoclonal PCs), and threshold, including sufficient information to determine the detection and quantification limits (LOD and LLOQ) of the individual assay (#86–95).

If the sample is not adequate, a possible note (e.g., hypocellular sample, peripheral blood contamination due to the low percentage of erythroblasts and/or myeloid precursors and/or mast cells and/or B-cell precursors) can be reported. According to the next-generation flow approach, a minimum of 1x10^7^ events is pivotal to support high-sensitive MRD-negative results; therefore, this goal must be declared in the report. An example of the final conclusion could be: ‘The plasma cell population—CD138 and/or CD38^+^—represents xx% of the entire cell population. The analysis conducted on the PC population documents xx% of clonal/pathological elements CDxx^pos^, CDxx^neg^, restricted for the Kappa/Lambda Ig light chains flanked by CD19/CD45^+^ PCs with a normal Kappa/Lambda ratio. The B lymphoid population, which represents xx% of lymphocytes, has a normal/clonal Kappa/Lambda expression. MRD study: negative, positive, below the limit of quantification’.

A report example is provided in Appendix A.

## 4. Discussion

The EuroFlow and IMWG have clearly defined the most informative markers and procedures in the characterization of MM, according to the next-generation flow approach. Less standardized is the way in which these data are reported, and no collaborative study has focused on a consensus for the flow cytometry clinical report shared by clinicians and laboratories. Specific competencies are required to provide a reliable and accurate evaluation in flow-MRD studies, and such difficulties are still limiting the consistency of MRD determination outside a few specialized laboratories. The progressive introduction of innovative, automated software for flow cytometry data analysis should significantly improve the standardization of such complex multiparametric analysis in order to provide accurate and reproducible MRD evaluations. However, operators’ experience remains essential for reliable interpretation of the results, especially in the most demanding cases [2,10], and it is extremely important that the clinician has a precise knowledge of the quality and reliability of the analytical process.

Several efforts of harmonization or standardization of flow MRD assessment have been performed, resulting in high concordance with regard to the presence or absence of MRD in inter-laboratory and inter-operator studies, supporting the use of the validated protocols in multicentre clinical trials [9,13,21,22]. “Defining the undetectable” is the current panorama of minimal residual disease evaluation in MM [10]. This real-life study, based on expert-shared knowledge and laboratory expertise and approved by clinicians, aims to ensure the operational autonomy of each laboratory regardless of the analysis software, reagents and flow cytometer available, provided that they are validated and approved for in vitro laboratory diagnostic procedures. The aim was to identify the necessary data to be included in the clinical report for multicentre application in MM diagnosis and MRD reporting, to facilitate data collection, to provide the clinician with a report that limits subjective interpretations, underlines possible limitations of the study, and helps to guide the clinical decision-making process.

The study was planned into 3 different parts: (a) the pre-analytical phase, (b) the analytical phase, and (c) the final conclusion. Only the points of the checklist agreed upon by ≥75% of the participant have been included in the final report. The pre-analytical phase focuses on patient personal information, clinical data and characteristics of the biological material to be processed, highlighting the relevance of the first pool and timing of sample processing, as well as any new innovative treatment which could interfere with PC detection.

Adequate collection of bone marrow samples is of primary importance to ensure reliable quantification of the rare event analysis. According to the 2017 European Leukemia Net guidelines for acute myeloid leukemia, the collection of a small volume of bone marrow (3–5 mL) is recommended [16]. EuroFlow recommends a median volume of 1.5 mL of BM for MM flow-MRD studies [23]. Consequently, as a quality control of the sampling procedure, it was agreed to report the amount (ml) and timing of the blood collection since the first pool is the most suitable tissue for MRD studies [2,16,17,24], as well as the absolute number of white blood cells of the sample as assessed by an automatic blood cell counter. Appendix A.

CD138 (syndecan-1) is a canonical PC marker. However, the CD138 intensity of expression can significantly decrease, up to disappearing, on the PC membrane in a few days after bone marrow sample collection, causing a reduction in the sensitivity of PC identification through the use of this marker. Timely sample processing is an essential pre-requisite for the high validity of flow-MRD studies, and IMWG recommends processing MM flow-MRD samples within 24–48 h. This requirement poses challenges for the use of single reference labs in multinational trials due to shipping delays and emphasizes a need for local testing; thereafter, reporting the date of sample processing was strongly recommended.

Unlike other onco-hematological malignancies, such as acute myeloid leukemia, a major antigenic shift of pathological PCs during treatment is a rare event [24]. However, treatment with the anti-CD38 monoclonal antibodies [25,26], recently approved in first-line treatment, masks PC epitopes overlapping with all standard diagnostic anti-CD38 monoclonal antibodies. This may interfere with CD38 detection by flow cytometry for several months after the last antibody infusion, thus representing a diagnostic challenge [27,28]. To bypass this technical problem, it has been proposed to include alternative markers such as multi-epitope CD38, VS38c, CD319, CD200 or others in the panel for the identification of PCs [29,30,31]. All participants agreed on the need to specify in the report whether anti-CD38 treatment was performed, as well as whether markers other than CD38 and CD138 were used for PC identification.

The analytical phase points out the quality of the sample, the analysis of PC and lymphoid populations, and the way to report the results.

Evaluation of sample quality is critical and a priority after treatment, with poor-quality bone marrow samples being the major cause of false negative results. Flow morphology is a highly reliable strategy for defining and documenting sample quality. In line with the current guidelines, the inclusion of a cytometric myelogram in the report is proposed through the use of the recommended markers for PC characterization. Appendix A. Bone marrow-specific cell type and reference values have been provided by EuroFlow studies, documenting that greater specificity in defining hemodilution samples could potentially be achieved by combining the complete or simultaneous absence of well-defined subpopulations [22]. Erythroblasts, precursor B-cells and mast cells are considered reliable warrantors of a bone marrow sample. Thereafter, their percentage should be included in the report, as a decrease/absence of these subpopulation is a possible index of hemodilution.

MM is complicated in 4–10% of cases by myelodysplastic syndromes (MDS) [32]. Although a low incidence in untreated MM patients has been reported [33], the next-generation sequencing approach is documented to greatly help with the diagnosis of myeloid neoplasms in MM [34,35]. MDS and acute myeloid leukemia (AML) can be late complications following mutagenic treatment [36]. The increasing number of treatments available has significantly improved the survival of patients with MM. However, a number of agents and their cumulative doses have been significantly associated with an increased risk of developing therapy-related myeloid neoplasms [26,37,38,39,40,41]. Limited data are available on the outcome of patients developing MDS and AML after treatment for MM [42]. Thereafter, the early identification of patients at risk of developing MDS/AML is of the utmost importance. Reporting an increased percentage of CD117+ myeloid precursors evaluated as CD117 expression outside the PC population is strongly recommended and makes diagnostic investigations of MDS advisable.

Monocyte identification can represent some difficulties in myeloma due to their overlap with the PC population. Several strategies and related warnings have been reported, with CD45/SSC^int^ being the most used gate in this study. CD56 expression on monocytes has recently been correlated with a possible condition of myelodysplasia in MM [19] and may be useful in the distinction of myeloid neoplasm from reactive conditions [43,44]. However, a significant increase in the CD56^+^ monocyte population has been documented in patients with moderate and severe COVID-19 [45,46] and has been positively correlated with body mass index [47]. Adequate clinical integration with cytometric data is therefore strongly recommended before undertaking further investigation regarding a possible MDS in patients with myeloma. Only 50% of the participants considered it appropriate to include the percentage of CD56-positive monocytes in the report, as further studies are needed to confirm the clinical role of this observation.

PC were identified based on their unique pattern of expression of CD38, CD138, and CD45 and light scatter features following consensus recommendations. The percentage of pathological cells out of the total leukocyte population, assessed by flow cytometry, is of primary importance in the diagnosis of acute leukemia and myelodysplastic syndrome. In contrast, the percentage of PC documented by cytometry is several logarithms lower than the morphology. To highlight the PC population, elevating it from the vast majority of non-informative events (myelocytes, granulocytes and lymphocytes), it was agreed to report the percentage of both clonal and normal elements within the PC population. To produce a simpler and more immediate understanding of the study, the results are reported, focusing on the target population. Appendix A.

In the monitoring of the disease, the temporal factor has recently gained great importance. If a positive MRD is a negative prognostic factor, a negative MRD must be sustained over time to acquire clinical relevance [11,48,49,50,51]. Consequently, the trend over time of the proportion between normal and clonal PCs may have prognostic relevance [17]. Furthermore, other factors, such as a protective immune profile, may explain the prolonged survival observed in some cases despite persistent MRD [52]. The *ratio* of the percentage of clonal cells to total PCs can be a data normalizer; it is not affected by hemodilution, reducing the significant prevalence of non-informative events (e.g., granulocytes). However, only 62% of the laboratories agreed to include the *ratio* of clonal to total plasma cells in the clinical report. A cooperative study is underway to evaluate the dynamic change in monoclonal/polyclonal PC balance at follow-up after treatment.

In PC analysis, no single parameter reliably distinguishes clonal from normal PCs, and the identification of neoplastic PCs should never be based on a single SAM. Moreover, a minor subset of normal PCs, in percentages that can reach 30% of the total PC population, can express antigens that characterize a pathological population (CD45^neg^, CD19^neg^, or CD56^+^) [41,53]: consequently, the analysis of Ig light chains expression within the PC subpopulations remains a valuable tool to obtain a clear-cut distinction between clonal and normal PCs [54,55]. Furthermore, the tumor clone at diagnosis often exhibits phenotypic heterogeneity, making it advisable to follow all the different phenotypic subclones during post-treatment monitoring. All participants agreed that the Ig light chain Kappa/Lambda *ratio* on PC subpopulations represents a key marker for discriminating between tumor and normal PCs (pathological Kappa/Lambda ratio <0.5 or >4.0), and this value must be reported. It is recommended to evaluate the Kappa/Lambda expression on each informative antigen: both on the markers that identify a normal population (e.g., CD19/CD45^+^, and CD38/CD138^bright^) and on the SAM, alone and in positive and negative combinations with ‘normal’. In combination with a bright CD45/CD19^+,^ PCs represent, in most cases, a normal population. Consequently, a normal Kappa/Lambda *ratio* in this subpopulation provides a strong internal quality control; therefore, the polyclonal PC levels should always be stated in the final report. However, it should be taken into consideration that a minority of myelomas are CD19^+^ and that this marker can therefore be expressed by neoplastic PCs at onset and thereafter.

Despite the presence in the panel of a considerable number of markers describing the lymphoid population (B-cell precursors, immature, naïve, memory, T, and NK cells), only the percentage of CD19/CD20^+^ B lymphocytes was included in the report. In PC dyscrasia, the restriction of the expression of Ig light chains on the lymphocyte population could suggest a diagnosis of Waldenstrom macroglobulinemia, making it appropriate to expand the phenotypic study consequently [56]. Furthermore, several laboratories involved in this study have experienced the coexistence of a clonal B lymphoproliferative disease (e.g., chronic lymphocytic leukemia and hairy cell leukemia) together with MM in a minority of cases. Therefore, it is strongly recommended to report both the *ratio* of Ig light chains of the lymphoid population and to expand the lymphocyte characterization panel in case of clonal restriction of B lymphocytes. Furthermore, the expression of Ig light chains on CD38^+^ and CD38^−^ B lymphocytes identify the B-cell precursors (CD38/CD81^+^, no Ig light chain expression) and the mature B-cells (CD38^-^ with a balanced Kappa/Lambda *ratio*), representing a reliable quality control of the intra-cytoplasmic staining. Appendix A.

While MRD-positive results are informative, negative results may be false or misleading, and a high degree of attention to several aspects is required facing an MRD-negative report. The current landscape of MRD testing in MM is characterized by the strengths and weaknesses of the various methods, and there are many open questions about the current testing procedures. The application flow-MRD measurement is not yet a routine clinical practice in MM. However, flow cytometry testing can reach a sensitivity similar to a molecular approach, such as a real-time polymerase chain reaction (≤10^−5^).

To perform rare events analysis, the number of events to acquire and the number of events representing the target population is of primary relevance [57]. In order to establish the detection sensitivity to enumerate a low number of true PCs from the background, the limit of blank (LOB), limit of detection (LOD), and lower limit of quantification (LLOQ) must be established [53,58]. Depending on the method used, the minimum number of abnormal cells required for an MRD-positive result varies from 20 to 50 clonal PCs [10]. The most sensitive flow cytometry techniques available include the EuroFlow recommendations for an 8-color, 2-tube panel with a reported sensitivity of 2 tumor cells in 1,000,000 (10^−6^) cells, given that the recommended 10 million cells are assayed, and the U.S. version, developed by the Memorial Sloan Kettering Cancer Center in New York, based on a 10-color single tube panel with a sensitivity of 6 tumor cells in 1,000,000 provided that 3 million can be assayed [59]. To date, MRD negativity has been shown to confer PFS and OS benefits through a sensitivity threshold of 10^−6^. Further evaluations of different tests and cut-offs are needed in the face of an ever-changing spectrum of diagnostic tools that detect progressively smaller quantities of disease. In our consensus work, in reporting MRD status, each laboratory should include the sensitivity achieved (10^−4^/10^−5^/10^−6^) and the MRD status: negative, positive or “below the limit of quantification” if the number of clonal PCs is between the LOD (≥20) and LLOQ (≤50).

The harmonization process utilized in this study uses all possible internal quality controls to validate the flow cytometry result. The keywords are: sample control, process control, and clarity of conclusion. In this regard, the United Kingdom National External Quality Assessment Service (UK NEQAS) and EuroFlow are developing innovative external quality programs which aim at monitoring the flow cytometry process for MM MRD testing that will allow a better inter-laboratory alignment [21]. Innovative transversal external quality controls are strongly needed for the analysis of rare events in the context of in vitro diagnostic (IVD) products; they will provide valuable support to the application of flow-MRD in routine clinical practice.

## 5. Conclusions

The process of achieving harmonization of the report may seem difficult; however, the conclusion is a short and clear sentence that states whether the biological sample is suitable for the study and whether a clonal PC population is present or absent. All laboratories agreed on the extreme care that must be taken in providing an MRD-negative result, as a positive MRD result is highly reliable in MM analysis if supported by an adequate number of informative events.

A possible limitation in the application of this type of report may be due to the reporting system available in the laboratory. The development of innovative and certified computer programs dedicated to data reporting and disease monitoring in diagnostic cytometry is ongoing and strongly needed.

## Figures and Tables

**Figure 1 cancers-15-02060-f001:**
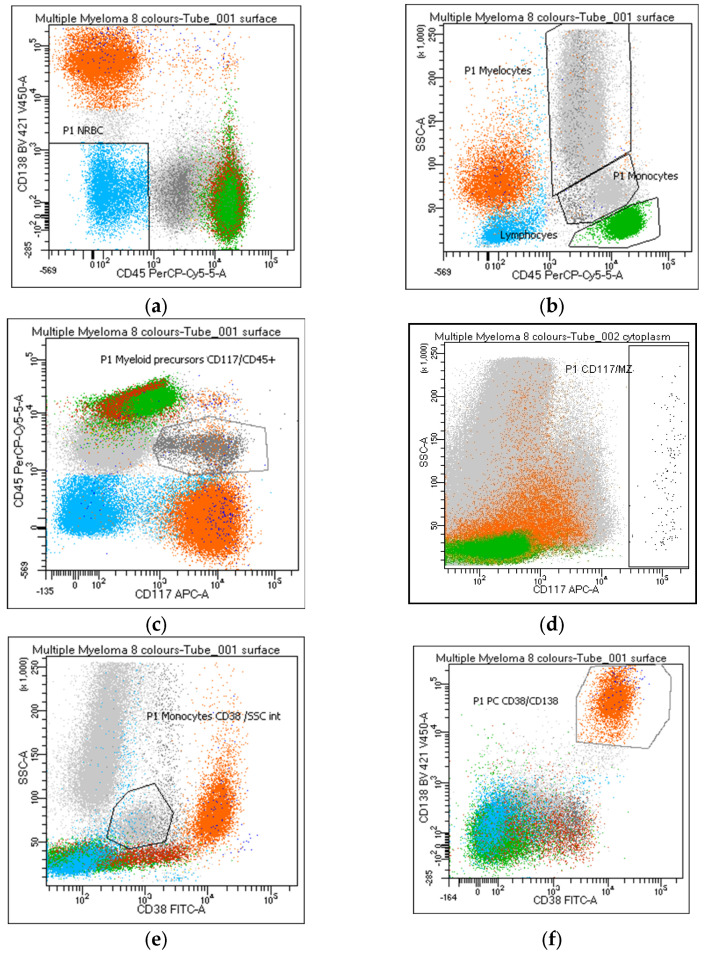
Identification of the distinct bone marrow populations by flow cytometry panel routinely used for MM diagnosis and monitoring. White blood cells were identified in the dot plot and enumerated according to the following criteria: (**a**) nucleated red blood cells—NRBC: CD45^neg^/CD138^neg^ (azure); (**b**) lymphocytes: CD45^++^/SSC^low^ (green); (**b**) myelocytes: CD45/SSC^hi^ (grey); (**c**) myeloid precursors: CD117/CD45^+^ (grey); (**d**) mast cells: CD117^hi^/SSC^hi^ (black); (**b**,**e**) monocytes: CD45 or CD38/SSC^int^ (grey); (**f**) plasma cells: CD38/CD138^++^ (orange).

**Figure 2 cancers-15-02060-f002:**
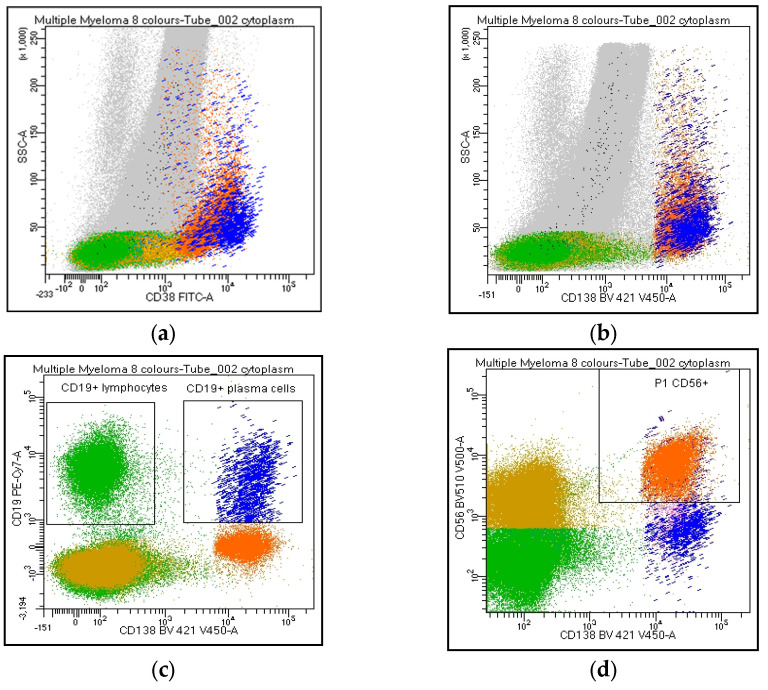
Flow cytometry identification of Kappa/Lambda Ig light chains expression on plasma cell and lymphoid population. The figure shows Ig light chains expression on CD38 and/or CD138^bright^ PCs (**a**,**b**, orange), on CD19^+^/CD138^neg^ lymphocytes (**c**, green), CD19/CD138^+^ (**c**, blue), and CD56/CD138^+^ (**d**, orange) PCs. Figure 2e describes a balanced Kappa/Lambda ratio on CD19^+^ PCs (blue), clonal Kappa/Lambda ratio on CD56^+^ PCs (orange), and a balanced Kappa/Lambda ratio on CD19^+^ lymphocytes (green).

**Table 1 cancers-15-02060-t001:** Patient personal information, clinical data and biological materials.

**#**		Y	N	NR
1	Name	8/8		
2	Sex	7/8		1/8
3	Date of birth	8/8		
4	Code of sample identification	8/8		
5	Medical center of origin	8/8		
6	Requesting physician	8/8		
7	Diagnostic query	8/8		
8	Timepoint of assessment/stage	8/8		
9	Type of biological material (bone marrow, peripheral blood	8/8		
10	Type of anticoagulant	3/8		5/8
11	Amount of biological material (ml)	6/8		2/8
12	First pool (for MRD studies)	7/8		1/8
13	Cellularity/mm^3^ (from automatic cell counter)	6/8		2/8
14	Date of sampling	8/8		
15	Date of processing	8/8		
16	Date of report	8/8		
17	Treatment (to specify MoAb treatment such as anti-CD38)	8/8		

**Table 2 cancers-15-02060-t002:** List of information regarding the methods used and the number of events analyzed.

#		Y	N	NR
18	Bulk lysis (yes/no)	4/8	1/8	3/8
19	Cell fixation and permeabilizing reagents for intra-cytoplasmic staining	1/8		7/8
20	Gating strategy	7/8		1/8
21	Number of total events acquired/analyzed	8/8		
22	Number of plasma cells acquired/analyzed	8/8		

**Table 3 cancers-15-02060-t003:** Quality of the sample: identification of the distinct BM populations by antibody panel routinely used for MM diagnosis and monitoring with relative warning/potential pitfalls of the analysis.

**#**	**Cell** **Population %**	**Markers**	**Warning/Potential Pitfalls**	Y	N	NR
23	NRBC	CD45^neg^/CD38 and or CD138^neg^	If reduced, possible risk of hemodilution	7/8		1/8
24		CD45^neg^/SSC^low^	NRBC overestimation if PCs CD45^neg^/small size	5/8		3/8
25		NRBC value from the WBC count		1/8	1/8	6/8
26	Lymphocytes	FSC/SSC		6/8		2/8
27		SSC^low^/CD45^++^		6/8		2/8
28	Hematogones	CD19/CD38^++^ or CD81^++^ lymphocytes	If reduced, possible risk of hemodilution	6/8		2/8
29	Myelocytes	SSC^hi^/CD45^+^		6/8		2/8
30	Myeloid precursors	CD117/CD45	CD45 CD117+ PCs included	7/8		1/8
31		CD117^+^/CD138^neg^		7/8		1/8
32		CD117/CD45 > 5%	Marker of putative myelodysplasia	7/8		1/8
33	Mast cells	CD117hi/SSChi	If reduced, possible risk of hemodilution	6/8		2/8
34	Monocytes	FSC/SSC	PCs can be included	5/8		3/8
35		CD45/SSC^int^	CD45^+^ PCs included	6/8		2/8
36		CD38^+^/SSC^int^		5/8		3/8
37		CD56 on CD38/SSC^int^	If >50% marker of putative myelodysplasia	4/8		4/8
38	Plasma cells	CD38^++^/SSC	Exclude CD38^neg^ MM	7/8		1/8
39			PCs negative after anti-CD38 therapy	8/8		
40		CD138^++^/SSC	Rare CD138^dim^/^neg^ MM	8/8		
41		CD38/CD138^++^	CD38 or CD138^neg^ PCs excluded	8/8		
42		CD38 multi-epitope or VS38c/other markers		8/8		

**Table 4 cancers-15-02060-t004:** Phenotypic analysis on PC and lymphoid populations.

**#**			Y	N	NR
43	List of the antibodies tested		6/8	1/8	1/8
44	Percentage/number of plasma cells on total cellularity		8/8		
45	Percentage/number of neoplastic cells on total cellularity		5/5		3/8
46	Percentage of normal plasma cells on total cellularity		5/8	1/8	2/8
47	Percentage/number of neoplastic cells within the plasma cell population		7/8		1/8
48	Percentage of normal plasma cells within the plasma cell population		7/8		1/8
49	Ratio clonal PCs/total PCs		5/8		3/8
* **Markers to identify the PC population** *					
50	CD38^+^ bright		8/8		
51	CD138^+^ bright		8/8		
52	CD38/CD138^+^ bright		8/8		
53	CD38 multi-epitope, VS38c, CD319 or other markers		8/8		
* **Analysis on plasma cell (CD38 and/or CD138^bright^) population (%)** *					
54	CD19		8/8		
55	CD20		8/8		
56	CD27		7/8		1/8
57	CD28		7/8		1/8
58	CD45		8/8		
59	CD56		8/8		
60	CD81		6/8		2/8
61	CD117		8/8		
62	Cytoplasmic Ig Kappa light chains		8/8		
63	Cytoplasmic Ig Lambda light chains		8/8		
64	CD19^+^/CD45^+^	Putative polyclonal PCs on CD19^neg^ MM	5/8		3/8
65	CD19^+^/SAM^+^	Putative monoclonal PCs on CD19^+^ MM	6/8		2/8
66	CD19^+^/SAM^neg^	Putative polyclonal PCs on CD19^neg^ MM	6/8		2/8
67	CD19^neg^/SAM^+^	Putative monoclonal PCs	6/8		2/8
68	Ig Kappa/Lambda ratio on	Total PCs	6/8		2/8
69		CD19^+^ PCs	6/8		2/8
70		SAM^+^ PCs	6/8		2/8
71		CD19^neg^/SAM^+^ PCs	6/8		2/8
72		CD19^+^/SAM^neg^ PCs	6/8		2/8
73		CD19^+^/CD45^+^ PCs	6/8		2/8
* **Analysis of lymphoid (SSC low/CD45++) population (%)** *					
74	CD19		8/8		
75	CD19+/CD38+		7/8	1/8	
76	CD19+/CD38neg		7/8	1/8	
77	CD20		8/8		
78	CD27		3/8	2/8	3/8
79	CD28		3/8	2/8	3/8
80	CD56		4/8	2/8	2/8
81	CD81		3/8	2/8	3/8
82	Other		4/8	2/8	2/8
83	Ig Kappa/Lambda ratio on	CD19+	6/8	1/8	1/8
84		CD19+/CD38+	5/8	1/8	2/8
85		CD19+/CD38neg	4/8	2/8	2/8

**Table 5 cancers-15-02060-t005:** Final comment/conclusion of report.

#		Y	N	NR
86	State if clonal plasma cells are present or absent	8/8		
87	List of the aberrant markers expressed in the clonal plasma cell population	8/8		
88	List of the markers expressed in the normal plasma cell population	5/8	1/8	2/8
89	Quality of the sample: adequate, inadequate	8/8		
90	Maximum sensitivity of the method: 10^−5^/<0.001% (or the sensitivity reached)	8/8		
91	MRD status: MRD-positive; MRD-negative	7/8	1/8	
92	MRD threshold: 0.01%/10^−4^ or 0.001%/10^−5^	7/8	1/8	
93	MRD negative <20 clonal plasma cells MRD positive not quantifiable ≥ 20 <50 clonal plasma cells MRD positive ≥50 clonal plasma cells	6/8	1/8	1/8
94	Up-to-date reference for MRD	3/8	2/8	3/8
95	LOD (20 events) LLOQ (50 events)	8/8		

## Data Availability

Data supporting the conclusions of this study are available upon reasonable request to the corresponding author.

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
