# Peer review of "Consensus for Flow Cytometry Clinical Report on Multiple Myeloma: A Multicenter Harmonization Process Merging Laboratory Experience and Clinical Needs"

_cancers, 2023, doi:10.3390/cancers15072060_

Round 1

Reviewer 1 Report

The manuscript entitled Consensus for Flow Cytometry Clinical Report on Multiple Myeloma: A Multicenter Harmonization Process Merging Laboratory Experience and Clinical Needs” is a detailed and practical guideline on how to report immunophenotyping data in MM including important aspect of MRD testing. The important additional issues of B-cell clonality reporting (coexistence of lymphoma) and myelodysplastic features (coexistence of MDS/AML) were also discussed. Although some points are repeated too often, overall, the paper is novel, useful and well written. 

I have only minor comments that could be considered and discussed by the Authors.

Minor comments.

1.     The Authors rightly emphasize the usefulness of reporting whether the bone marrow sample comes from the first pool as this information may help in resolving the issue of the quality of the material and thus the credibility of the result. However, although we can cytometrically determine certain leukocyte populations, the deficiency of which may indicate an admixture of blood, the reference ranges for them will depend, for example, on the time of the examination or the type of therapy. Please see and discuss e.g. Puig N, Flores-Montero J, Burgos L, et al. Reference Values to Assess Hemodilution and Warn of Potential False-Negative Minimal Residual Disease Results in Myeloma. Cancers (Basel). 2021;13(19).

2.     Drawing attention to the increased percentage of CD117+ myeloid precursors and the expression of CD56 on monocytes as possibly indicative of concomitant MDS is also of importance. The frequency of such findings as well as clinical outcome should be briefly discussed with appropriate literature cited if available. It may be also added that CD56 may be also expressed on reactive monocytes.

3.     The term “not quantifiable” referring to the situation of detecting between 20 and 50 myeloma cells should be better explained, also to be clear for clinicians ordering MRD tests.

4.     Literature should be carefully searched for other efforts of harmonization or standardization of flow MRD assessment in multiple myeloma that should be briefly discussed, e.g. KrzywdziÅ„ska A, et al. Harmonization of Flow Cytometric Minimal Residual Disease Assessment in Multiple Myeloma in Centers of Polish Myeloma Consortium. Diagnostics (Basel). 2021 

Author Response

Reviewer n°1

Point 1

Bone marrow specific cell type and reference values have been provided by EuroFlow studies, documenting that greater specificity in defining hemodilution samples could potentially be achieved by combining the complete or simultaneous absences of three well defined subpopulations. Erythroblast, lymphocytes, precursors B-cell and mast cells are considered a reliable warrantor of a bone marrow sample. Thereafter, their percentage should be included in the report, being a high percentage of lymphocytes as well as a decrease/absence of erythroblast and/or hematogones and/or myeloid precursors and mast cells a possible index of peripheral blood contamination.

The following reference has been added and discussed.

  • Puig, N., Flores-Montero, J., Burgos, L., Cedena, M. T., Cordón, L., Pérez, J. J., Sanoja-Flores, L., Manrique, I., Rodríguez-Otero, P., Rosiñol, L., Martínez-López, J., Mateos, M. V., Lahuerta, J. J., Bladé, J., San Miguel, J. F., Orfao, A., & Paiva, B. (2021). Reference Values to Assess Hemodilution and Warn of Potential False-Negative Minimal Residual Disease Results in Myeloma. Cancers13(19), 4924. https://doi.org/10.3390/cancers13194924

Point 2a

MM is complicated in 4-10% of cases by myelodysplastic syndromes (MDS). Although a low incidence in untreated MM patients has been reported, the next generation sequencing approach is documenting to greatly help with myeloid neoplasms diagnosis in MM.

MDS and acute myeloid leukemia (AML) can be late complications following mutagenic treatment. The increasing number of treatments available has significantly improved the survival of patients with MM, however a number of agents and their cumulative doses have been significantly associated with an increased risk of developing therapy-related myeloid neoplasms. Limited data is available on the outcome of patients developing MDS and AML after treatment for MM. Thereafter, an early identification of patients at risk of developing MDS/AML is of upmost importance. Reporting of an increased percentage of CD117+ myeloid precursors evaluated as CD117 expression outside the PC population is strongly recommended, and makes diagnostic investigations of MDS advisable.

The following references have been added and discussed

  • Yang J, Terebelo HR, Zonder JA. Secondary primary malignancies in multiple myeloma: an old NEMESIS revisited. Adv Hematol. 2012;2012:801495
  • Mailankody S, Pfeiffer RM, Kristinsson SY, Korde N, Bjorkholm M, Goldin LR, Turesson I, Landgren O. Risk of acute myeloid leukemia and myelodysplastic syndromes after multiple myeloma and its precursor disease (MGUS) 2011;118:4086–4092.
  • Bolli N, Genuardi E, Ziccheddu B, Martello M, Oliva S, Terragna C. Next-Generation Sequencing for Clinical Management of Multiple Myeloma: Ready for Prime Time? Front Oncol. 2020 Feb 25;10:189. doi: 10.3389/fonc.2020.00189. PMID: 32181154; PMCID: PMC7057289.
  • Song J, Zhang H, Dong N, Zhang X, Hussaini M, Jain A, Moscinski L, Shain K, Baz R, Alsina M, Nishihori T, Zhang L. The Application of NextGen Sequencing in the Diagnosis of Myeloid Neoplasms in Myeloma Patients With Cytopenia. Clin Lymphoma Myeloma Leuk. 2022 Jun;22(6):e414-e426. doi: 10.1016/j.clml.2021.12.008. Epub 2021 Dec 11. PMID: 34998786.
  • Arber DA, Orazi A, Hasserjian R, Thiele J, Borowitz MJ, Le Beau MM, Bloomfield CD, Cazzola M, Vardiman JW. The 2016 revision to the World Health Organization classification of myeloid neoplasms and acute leukemia. 2016;127(20):2391–2405. doi: 10.1182/blood-2016-03-643544.
  • Leone G, Pagano L, Ben-Yehuda D, Voso MT. Therapy-related leukemia and myelodysplasia: susceptibility and incidence. 2007;92(10):1389–1398. doi: 10.3324/haematol.11034.
  • Gertz MA, Terpos E, Dispenzieri A, Kumar S, Shah RA, Orlowski R, Kastritis E, Dimopoulos MA, Shah J. Therapy-related myelodysplastic syndrome/acute leukemia after multiple myeloma in the era of novel agents. Leuk Lymphoma. 2015 Jun;56(6):1723-6. doi: 10.3109/10428194.2014.970543. Epub 2014 Nov 3. PMID: 25284489.
  • Poh C, Keegan T, Rosenberg AS. Second primary malignancies in multiple myeloma: A review. Blood Rev. 2021;46:100757. doi: 10.1016/j.blre.2020.100757
  • Gudbjorg Jonsdottir, Magnus Björkholm, Ingemar Turesson, Malin Hultcrantz, Benjamin Diamond, Anna Porwit, Ola Landgren, Sigurdur Y. Kristinsson. Cumulative exposure to melphalan chemotherapy and subsequent risk of developing acute myeloid leukemia and myelodysplastic syndromes in patients with multiple myeloma. European Journal of Haematology 09 May 2021 https://doi.org/10.1111/ejh.13650
  • Boquoi A, Banahan SM, Mohring A, Savickaite I, Strapatsas J, Hildebrandt B, Kobbe G, Gattermann N, Haas R, Schroeder T, Germing U, Fenk R. Therapy-related myeloid neoplasms following treatment for multiple myeloma-a single center analysis. Ann Hematol. 2022 May;101(5):1031-1038. doi: 10.1007/s00277-022-04775-1. Epub 2022 Mar 9. PMID: 35262868; PMCID: PMC8993729.

Point 2b

CD56 expression on monocytes has recently been correlated with a possible condition of myelodysplasia in MM and may be useful in the distinction of myeloid neoplasm from reactive conditions. However, significant increase in the CD56+ monocyte population has been documented in patients with moderate and severe COVID-19 and has been positively correlated with body mass index. Adequate clinical integration with cytometric data is therefore strongly recommended before undertaking further investigation regarding a possible MDS in patients with myeloma. Only 50% of the participants considered it appropriate to include the percentage of CD56 positive monocytes in the report, as further studies are needed to confirm the clinical role of this observation.

The following references have been added and discussed.

  • Alhan, C., Westers, T. M., Cremers, E. M., Cali, C., Ossenkoppele, G. J., & van de Loosdrecht, A. A. (2016). Application of flow cytometry for myelodysplastic syndromes: Pitfalls and technical considerations.  Part B, Clinical cytometry90(4), 358–367. https://doi.org/10.1002/cyto.b.21333
  • Brown LE, Zhang D, Cui W. Flow Cytometric Analysis of Monocytes and Granulocytes May Be Useful in the Distinction of Myeloid Neoplasms from Reactive Conditions: A Single Institution Experience and Literature Review. Ann Clin Lab Sci. 2020 May;50(3):327-332. PMID: 32581021
  • Campana, S., De Pasquale, C., Sidoti Migliore, G., Pezzino, G., Cavaliere, R., Venanzi Rullo, E., Nunnari, G., Caramori, G., David, A., Bonaccorsi, I., Pollicino, T., Carrega, P., & Ferlazzo, G. (2022). Cutting Edge: Hyperinflammatory Monocytes Expressing CD56 Abound in Severe COVID-19 Patients. Journal of immunology (Baltimore, Md. : 1950)209(4), 655–659. https://doi.org/10.4049/jimmunol.2200021
  • Dutt TS, LaVergne SM, Webb TL, Baxter BA, Stromberg S, McFann K, Berry K, Tipton M, Alnachoukati O, Zier L, Ebel G, Dunn J, Henao-Tamayo M, Ryan EP. Comprehensive Immune Profiling Reveals CD56+Monocytes and CD31+ Endothelial Cells Are Increased in Severe COVID-19 Disease. J Immunol. 2022 Feb 1;208(3):685-696. doi: 10.4049/jimmunol.2100830. Epub 2022 Jan 5. PMID: 34987111.
  • Friedrich K, Sommer M, Strobel S, Thrum S, Blüher M, Wagner U, Rossol M. Perturbation of the Monocyte Compartment in Human Obesity. Front Immunol. 2019 Aug 8;10:1874. doi: 10.3389/fimmu.2019.01874. PMID: 31440251; PMCID: PMC6694869.

Point 3

While MRD-positive results are informative, negative results may be false or misleading and a high degree of attention to several aspects is required facing an MRD-negative report. The current landscape of MRD-testing in MM is characterized by the strengths and weaknesses of the various methods and there are many open questions with the current testing procedures. In this regard, the difficulties and possible pitfalls in performing flow-MRD studies is underlined by the clinical relevance of a sustained MRD negative results. To perform rare events analysis, several parameters, including the total number of cells to acquire and the minimum number of events representing the target population have a great effect on sensitivity thresholds. In order to establish the detection sensitivity to enumerate low number of true PCs from the background, the limit of blank (LOB), limit of detection (LOD) and lower limit of quantification (LLOQ) must be established. Depending on the method used, the minimum number of abnormal cells required for an MRD-positive result varies from 20 to 50 clonal PCs. The most sensitive flow cytometry techniques available include the EuroFlow recommendations for an 8-color, 2-tube panel with reported sensitivity of 2 tumor cells in 1,000,000 (10−6) cells given the recommended 10 million cells are assayed and the U.S. version, developed by the Memorial Sloan Kettering Cancer Center in New York, based on a 10-color single tube panel with a sensitivity of 6 tumor cells in 1,000,000 provided 3 million can be assayed. Up to date, MRD-negativity has been shown to confer PFS and OS benefits though a sensitivity threshold of 10−6. Further evaluations of different tests and cut-offs are needed in the face of an ever-changing spectrum of diagnostic tools that detect progressively smaller quantities of disease. In our consensus work, in reporting MRD status each laboratory should include the sensitivity achieved (10-4/10-5/10-6) and the MRD status: negative, positive or "below the limit of quantification” when the number of clonal PC is between the LOD and LLOQ (20-50 abnormal PC).

The following references have been added and discussed.

  • Diamond BT, Rustad E, Maclachlan K, Thoren K, Ho C, Roshal M, Ulaner GA, Landgren CO. Defining the undetectable: The current landscape of minimal residual disease assessment in multiple myeloma and goals for future clarity. Blood Rev. 2021 Mar;46:100732. doi: 10.1016/j.blre.2020.100732. Epub 2020 Jul 10. PMID: 32771227; PMCID: PMC9928431.
  • Roshal M, Flores-Montero JA, Gao Q, Koeber M, Wardrope J, Durie BGM, et al., MRD detection in multiple myeloma: comparison between MSKCC 10-color single-tube and EuroFlow 8-color 2-tube methods. Blood Advances, 2017. 1(12): p. 728–732.

Point 4

Several efforts of harmonization or standardization of flow MRD assessment have been performed, resulting in high concordance with regard to the presence or absence of MRD in inter-laboratory and inter-operator studies, supporting the use of the validated protocols in multi-center clinical trials. However, the experience of operators remains essential for a reliable interpretation of the results, especially in more demanding cases.

The following references have been added and discussed.

  • Gozzetti A, Raspadori D, Bacchiarri F, Sicuranza A, Pacelli P, Ferrigno I, Tocci D, Bocchia M. Minimal Residual Disease in Multiple Myeloma: State of the Art and Applications in Clinical Practice. J Pers Med. 2020 Sep 10;10(3):120. doi: 10.3390/jpm10030120. PMID: 32927719; PMCID: PMC7565263.
  • KrzywdziÅ„ska A, PuÅ‚a B, Czyż A, Krzymieniewska B, Kiernicka-Parulska J, Mierzwa A, Szymczak D, Milanowska A, Kiraga A, KwiecieÅ„ I, Zaleska J, Jamroziak K. Harmonization of Flow Cytometric Minimal Residual Disease Assessment in Multiple Myeloma in Centers of Polish Myeloma Consortium. Diagnostics (Basel). 2021 Oct 11;11(10):1872. doi: 10.3390/diagnostics11101872. PMID: 34679569; PMCID: PMC8534749.

Reviewer 2 Report

Cordone performed a multicenter effort for harmonization of analysis and reporting MRD in myeloma by flow cytometry. Such studies are obviously useful given the increasing importance of MRD testing in MM. However, the paper is sometimes hard to follow for the average physician and should be improved by revising several parts and changing the flow of tables. 

Major comments

1. Tables should follow the corresponding text section for the sake of clarity and better understanding. For paradigm, table 2 has Y, N, NR and fractions of 8, but it is not readily understandable what NR stands for and what is 8.

2. Tables should be embedded as tables not figures.

3. Table 1 should go as a supplement.

4. Table 4 is incomprehensible.  First, the title “quality control of the sample” appears not to correspond to what this table presents. Second, it is not clear what are the warnings/pitfalls presented, how they were selected or what the answers are about? I.e., “if included in PC panel” seems not to be a warning, neither a pitfall. The authors should revise both the legend, the table and the corresponding text in order to correctly describe this stage of process (analysis strategies?)

5. In figure 1 the legend should inform on colours of dot plots, otherwise confuses the reader.

6. How was the 75% threshold chosen? Justification of this cut-off is important for the reliability of the final consensus

7. There is no reporting/consensus on the analysis software and cytometer type. These are generally taken into consideration when comparing FC results. Can the authors explain why they did not question about these two factors?

8. Discussion is pretty long and has many repeats, e.g. in lines 266-270, it is stated twice the real-life setting. It should be shortened and be more concise.

9. In lines 64-5, the citations (9-11) do not support the sentence (MRD-guided therapy).

10. The paper needs some English polishing, both for grammar (e.g. line 59, has making should change to has made) and syntax errors (e.g. abstract, lines 36-37 are rather incomprehensible).

Author Response

Reviewer n°2

Point 1

In the revised version, tables now follow the corresponding text section.

A comparison between the 8 laboratories regarding the strategies of phenotypic analysis as well as an agreement on the data that must be included in the final report was reached through the use of a questionnaire; yes (Y), no (N) or not relevant (NR) were the 3 possible answers.

Point 2

Tables have been embedded as tables and not figures.

Point 3

Table 1 has been included in the supplementary data.

Point 4

Table 4 shows the strategies to identify different bone marrow cells and their relative distribution, the so called “cytometric myelogram”. The presence and amount of key populations, usually absent in the peripheral blood, are utilized to assess the quality of the sample and eventually highlight/suggest hemodilution. The legend, the table and the corresponding text has been revised in order to better describe this stage of the process. CD33 has been excluded from the table.

Point 5

Figure 1: the colours of dot plot have been added.

Point 6

This real-life study, based on expert-shared knowledge and laboratory expertise and approved by clinicians, aims to share a flow cytometry clinical report for multicenter application in MM diagnosis and MRD reporting to assure accurate and sensitive assessment.

The points of the questionnaire focus on open questions and derive from the evaluation of the various reports used by the study participants. The cut-off of 75% was chosen, corresponding to an agreement of 6 out of 8 laboratories, to ensure the operational autonomy of each laboratory limiting any exclusion.

Point 7

The work aims to ensure the operational autonomy of each laboratory, regardless of the analysis software and flow cytometer available, provided that they are validated and approved for use in vitro laboratory diagnostics. Several efforts of harmonization or standardization of flow MRD assessment have been performed, resulting in high concordance with regard to the presence or absence of MRD in inter-laboratory and inter-operator studies, supporting the use of the validated protocols in multi-center clinical. However, the experience of operators remains essential for a reliable interpretation of the results, especially in more demanding cases.

The following references have been added and discussed.

  • Gozzetti A, Raspadori D, Bacchiarri F, Sicuranza A, Pacelli P, Ferrigno I, Tocci D, Bocchia M. Minimal Residual Disease in Multiple Myeloma: State of the Art and Applications in Clinical Practice. J Pers Med. 2020 Sep 10;10(3):120. doi: 10.3390/jpm10030120. PMID: 32927719; PMCID: PMC7565263.
  • KrzywdziÅ„ska A, PuÅ‚a B, Czyż A, Krzymieniewska B, Kiernicka-Parulska J, Mierzwa A, Szymczak D, Milanowska A, Kiraga A, KwiecieÅ„ I, Zaleska J, Jamroziak K. Harmonization of Flow Cytometric Minimal Residual Disease Assessment in Multiple Myeloma in Centers of Polish Myeloma Consortium. Diagnostics (Basel). 2021 Oct 11;11(10):1872. doi: 10.3390/diagnostics11101872. PMID: 34679569; PMCID: PMC8534749.

“Defining the undetectable” is the current panorama of minimal residual disease evaluation in MM (Diamond 2021). This real-life study, based on expert-shared knowledge and laboratory expertise and approved by clinicians, aims to share a flow cytomery clinical report for multicenter application in MM diagnosis and MRD reporting to assure accurate and sensitive assessment.

  • Diamond BT, Rustad E, Maclachlan K, Thoren K, Ho C, Roshal M, Ulaner GA, Landgren CO. Defining the undetectable: The current landscape of minimal residual disease assessment in multiple myeloma and goals for future clarity. Blood Rev. 2021 Mar;46:100732. doi: 10.1016/j.blre.2020.100732. Epub 2020 Jul 10. PMID: 32771227; PMCID: PMC9928431.

Point 8, 9 and 10

The discussion has been shortened and corrected. The citations have been verified and new references have been added. Grammatical and syntax errors have been corrected.

Reviewer 3 Report

The authors present a well written manuscript describing the important findings that should be reported for flow cytometry clinical report on MM MRD. The topic is interesting and the manuscript will hopefully lead to more uniform reporting.  

The manuscript can be improved by proving an example of an ideal report.  Appropriate sections of the example report can be incorporated into the discussion with each section preceding the appropriate paragraphs from the discussion.

Author Response

Reviewer n°3

An example of an ideal repost has been provided into the text and as supplementary table 2.

Patient personal information, clinical data and biological material

Family name

Name

Date of birth

Sex

Ward / clinic / hospital

Physician

Request ID

Report number

Sample type

First pull

yes

no

Sample ml

Sample cellularity

Date of collection

Date of cytometry study

Diagnostic query and clinical information

Quality control of the sample: cytometric myelogram

Erythroblasts

(CD138 neg / CD45 neg) = %

Lymphocytes

(SSC low / CD45++) = %

Myelocytes

(SSC high / CD45+) = %

Monocytes

(CD38/CD45/SSC intermediate) = %

Myeloid Precursors

(CD117/45+) = %

Mast Cells

(SSC high / CD117 high) = %

Plasma cells (CD138/38++) = %

N° of total events acquired =

N° of total plasma cells acquired =

Flow cytometry analysis of plasma cells and lymphoid population

Analysis on plasma cell population (CD38 / CD138 bright)

Marker %

Marker

Cyto Ig

Kappa %

Cyto Ig

Lambda %

Ratio kappa/lambda

CD38 = 100%++

CD38 positive

CD138 = 100%++

CD19 positive

CD19 = %

CD19 negative

CD20 = %

CD56 positive

CD27 = %

CD56 negative

CD28 = %

CD117 positive

CD56 = %

CD19 positive / CD45 positive

CD45 = %

CD19 positive / SAM positive

CD81 = %

CD19 positive / SAM negative

CD117 = %

CD19 negative / SAM positive

SAM = Surface Aberrant Marker; Abnormal Kappa / Lambda = <0.5 o >4.0

Analysis on lymphoid population (SSC/CD45++)

Marker %

Marker

Cyto Ig

Kappa %

Cyto Ig

Lambda %

Ratio kappa/lambda

CD19 = %

CD19 positive

CD20 = %

CD19 positive / CD38 negative

CD28 = %

CD19 positive / CD38 positive

CD56 = %

CD45 = %

CD19/CD38 = %

Final comment/conclusion

‘The plasma cell population - CD138 and/or CD38+ - represents xx% of the entire cell population. The analysis conducted on the PC population documents a xx% of clonal/pathological elements CDxxpos, CDxxneg, restricted for the Kappa/Lambda Ig light chains flanked by CD19/CD45+ PCs with a normal Kappa/Lambda ratio. The B lymphoid population, which represents xx% of lymphocytes, has a normal/clonal Kappa/Lambda expression. MRD study: negative, positive, below the limit of quantification’.

For MRD study: negative, positive, below the limit of quantification.

Unsuitable bone marrow sample: Hypocellular sample, possibly contaminated with peripheral blood due to low percentage of erythroblasts and/or myeloid precursors and mast cells and/or B-lymphoid precursors.

Reviewer 4 Report

In the manuscript “Consensus for Flow Cytometry Clinical Report on Multiple Myeloma: A Multicenter Harmonization Process Merging Laboratory Experience and Clinical Needs” the authors present the consensus reached by a group of laboratory experts and clinicians about the necessary data to be included in the clinical report of multiple myeloma studies by flow cytometry at diagnosis or after treatment.

General comments

The manuscript presents the analysis of a multiple choices questionnaires on each single stage of the process that was filled by 12 haematological Centers. An agreement on the data that must be included in the final report was reached through the use of a questionnaire and the discordant results were discussed collectively. However, although the experience of a local society might be of interest, there are no experimental results in this original manuscript that back up the experts opinion.

Based in previous reports, and authors experience, the manuscript suggest that there are several point of the pre-analytical and post-analytical phase that might impact the reproducibility of the results. However, no new information is provided about this. I do not know whether the authors have shared samples between centers or make experiments to check which prenalytical factors might impact the reproducibility of the results, and to evaluate the impact of these factors (eg. time from sampling, storage temperature, instrument setup protocol, speed of acquisition).

The same in true for the post analytical phase. There are specific recommendations that include a list of markers that need to be reported and a gating strategy. These recommendations are base again in experience and previous publications but not evidences are provided about the convenience of these. Seom of these post-analytical factor could be evaluated by a multicenter study, sharing large series of datafiles that provide objective information about the impact of these factors.

Author Response

Reviewer n°4

The author did not have shared samples between Centres.

Several efforts of harmonization or standardization of flow MRD assessment have been performed. The applicability and feasibility of the validated protocols has been already confirmed by multicenter study as highly sensitive method of MRD evaluation in MM. Moreover, harmonization of MRD assays resulted in high concordance with regard to the presence or absence of MRD in inter-laboratory and inter-operator studies supporting the use of the method in multi-center clinical trials. However, the experience of operators remains essential for a reliable interpretation of the results.

The work focuses on the "real life" experience of laboratory involved in MM flow cytometry characterization. “Defining the undetectable” is the current panorama of minimal residual disease evaluation in multiple myeloma and the goal of our work is to use the clinical report as a powerful tool able to provide a clear conclusion of the flow cytometry study highlighting, at the same time, all the possible pitfalls of the process, for the best clarity of the result.

The following references have been added and discussed.

  • Gozzetti A, Raspadori D, Bacchiarri F, Sicuranza A, Pacelli P, Ferrigno I, Tocci D, Bocchia M. Minimal Residual Disease in Multiple Myeloma: State of the Art and Applications in Clinical Practice. J Pers Med. 2020 Sep 10;10(3):120. doi: 10.3390/jpm10030120. PMID: 32927719; PMCID: PMC7565263.
  • KrzywdziÅ„ska A, PuÅ‚a B, Czyż A, Krzymieniewska B, Kiernicka-Parulska J, Mierzwa A, Szymczak D, Milanowska A, Kiraga A, KwiecieÅ„ I, Zaleska J, Jamroziak K. Harmonization of Flow Cytometric Minimal Residual Disease Assessment in Multiple Myeloma in Centers of Polish Myeloma Consortium. Diagnostics (Basel). 2021 Oct 11;11(10):1872. doi: 10.3390/diagnostics11101872. PMID: 34679569; PMCID: PMC8534749.

Round 2

Reviewer 2 Report

I have no further comments

Author Response

Thank you for your valuable suggestions.

Reviewer 3 Report

The authors have addressed my concerns.

Author Response

Thank you for your valuable suggestions

Reviewer 4 Report

The authors have not provided any additional experimental data. No substantial amount of new information is provided
